# PUFA stabilizes a conductive state of the selectivity filter in IKs channels

Alessia Golluscio[1,2], Jodene Eldstrom[3], Jessica J Jowais[1], Marta Elena Perez[1], Kevin Peter Cunningham[1,4], Alicia De La Cruz[1,2], Xiaoan Wu[1], Valentina Corradi[5], D Peter Tieleman[5], David Fedida[3], H Peter Larsson[1,2]*

[1]Department of Physiology and Biophysics, University of Miami, Miami, United States; [2]Department of Biomedical and Clinical Sciences, Linköping University, Linköping, Sweden; [3]Department of Anesthesiology, Pharmacology and Therapeutics, University of British Columbia, Vancouver, Canada; [4]School of Life Sciences, University of Westminster, London, United Kingdom; [5]Department of Biological Sciences and Centre for Molecular Simulation, University of Calgary, Calgary, Canada

## eLife assessment

This article reveals an **important** mechanism of KCNQ1/IKs channel gating and PUFA modulation of this mechanism. This mechanism is supported by **convincing** single-channel recordings, macroscopic current recordings, and mutational analyses. These findings are of importance to the ion channel field and possibly future therapeutic applications.

*For correspondence: peter.larsson@liu.se

**Abstract** In cardiomyocytes, the KCNQ1/KCNE1 channel complex mediates the slow delayed-rectifier current (IKs), pivotal during the repolarization phase of the ventricular action potential. Mutations in IKs cause long QT syndrome (LQTS), a syndrome with a prolonged QT interval on the ECG, which increases the risk of ventricular arrhythmia and sudden cardiac death. One potential therapeutical intervention for LQTS is based on targeting IKs channels to restore channel function and/or the physiological QT interval. Polyunsaturated fatty acids (PUFAs) are potent activators of KCNQ1 channels and activate IKs channels by binding to two different sites, one in the voltage sensor domain – which shifts the voltage dependence to more negative voltages – and the other in the pore domain – which increases the maximal conductance of the channels (Gmax). However, the mechanism by which PUFAs increase the Gmax of the IKs channels is still poorly understood. In addition, it is unclear why IKs channels have a very small single-channel conductance and a low open probability or whether PUFAs affect any of these properties of IKs channels. Our results suggest that the selectivity filter in KCNQ1 is normally unstable, contributing to the low open probability, and that the PUFA-induced increase in Gmax is caused by a stabilization of the selectivity filter in an open-conductive state.

## Introduction

The voltage-gated K⁺ channel KCNQ1, also referred to as Kv7.1, is expressed in the heart. Here the channel associates with the accessory KCNE1 subunit, generating the *so-called* slow delayed-rectifier (IKs) current, an important contributor to the repolarizing phase of the ventricular action potential (AP) (*Barhanin et al., 1996*; *Sanguinetti et al., 1996*; *Nerbonne and Kass, 2005*). Loss-of-function mutations of the KCNQ1/KCNE1 complex are associated with long QT syndrome (LQTS) (*Splawski et al., 2000*; *Tester and Ackerman, 2014*). These LQTS mutations cause a reduction of the channel current, leading to a significant prolongation of the ventricular AP waveform that can be seen on

**eLife digest** Travelling through the heart are waves of electrical activity that cause muscle cells to contract and pump blood around the body. The waves are generated by charged ions which flow via tiny channels in and out of the muscle cells. This electrical activity spreads quickly from one cell to the next to make sure all the muscle cells contract at the right time. When these ion channels are compromised, this can lead to heart problems such as long QT syndrome (LQTS).

In patients with LQTS, electrical activity in the heart does not follow the typical rhythm, which can result in an irregular heartbeat and lead to cardiac arrest. The most common cause of LQTS is mutations in the channel KCNQ1, which allows potassium ions to flow out of heart muscle cells. This outflux of potassium restores the electrical charge inside the cell so that it is ready to receive another electrical wave and contract at the right time.

Current treatments for LQTS do not target KCNQ1 channels directly and have side effects. An alternative approach could be to use a group of molecules called polyunsaturated fatty acids (or PUFAs for short) which increase the flow of ions that pass through KCNQ1. However, it is not fully understood how PUFAs achieve this.

Previous research showed that PUFAs activate KCNQ1 via two independent sites: one at the voltage sensor which decides whether the channel is open or closed (Site I), and another at the pore domain ions pass through (Site II). While it is well understood how PUFAs activate the channel at Site I, little is known about the activation mechanism that occurs at Site II.

To investigate, Golluscio et al. modified egg cells from the frog *Xenopus laevis* to express KCNQ1 channels. Experiments investigating the electrical properties of KCNQ1 revealed that the selective filter in the pore domain – which permits potassium but no other ions to pass through – is usually unstable. However, PUFAs help to stabilize this filter, causing KCNQ1 to stay open more often and allow potassium ions to flow out of muscle cells.

The findings of Golluscio et al. suggest that PUFAs could represent an important therapeutic tool to treat LQTS and potentially other cardiac disorders. However, further studies in heart cells, animals and eventually humans will be required to confirm this conclusion.

the electrocardiogram as a prolonged QT interval (*Nerbonne and Kass, 2005*; *Figure 1A*). These channel mutations, or dysfunction, increase the risk of developing cardiac arrhythmias, which can lead to sudden cardiac death (*Wu and Sanguinetti, 2016*). At present, treatment of LQTS is mainly based on the usage of β-blockers (most used are long-lasting preparations such as nadolol and atenolol) and implantable cardioverter-defibrillators, especially for patients with high risk of sudden death and frequent syncope (*Roden, 2008*). However, both approaches do not shorten the QT interval duration. Restoring the physiological duration of the QT interval could be achieved by increasing the activity of the KCNQ1/KCNE1 channel complex (*Varshneya et al., 2018*; *Figure 1*).

Like other Kv channels, KCNQ1 has a typical tetrameric structure of four α-subunits. Each α-subunit is composed of six transmembrane segments, with S1-S4 forming the voltage sensor domain (VSD) and S5-S6 forming the pore domain (PD) (*Figure 1D*). However, the KCNQ1/KCNE1 channel complex has a very small single-channel conductance and a very low open probability compared to other Kv channels (*Werry et al., 2013*). The mechanisms behind the small conductance and low open probability are not understood.

Polyunsaturated fatty acids (PUFA), in particular omega 3, are known to exert a protective effect on sudden cardiac death and are recommended in the diet at least twice a week (*Fernandez et al., 2021*). PUFAs have been shown to increase IKs currents by a dual mechanism of action, characteristically described as a 'Lipoelectric mechanism' (*Elinder and Liin, 2017*). According to this mechanism, PUFAs can increase IKs currents by shifting the voltage dependence of activation ($\Delta V0.5$) toward negative voltages and increase the maximum conductance of the channel ($\Delta Gmax$) (*Bohannon et al., 2019*; *Bohannon et al., 2020*; *Yazdi et al., 2021*; *Larsson et al., 2020*; *Wu and Larsson, 2020*; *Figure 1B and C*).

The two PUFA effects ($\Delta V0.5$ and $\Delta Gmax$) on KCNQ1 are independent of each other and originate from the binding of PUFA to two different sites, conventionally indicated as Site I and Site II (*Yazdi et al., 2021*; *Liin et al., 2018*). Site I is found in the VSD, where the PUFA head group interacts with

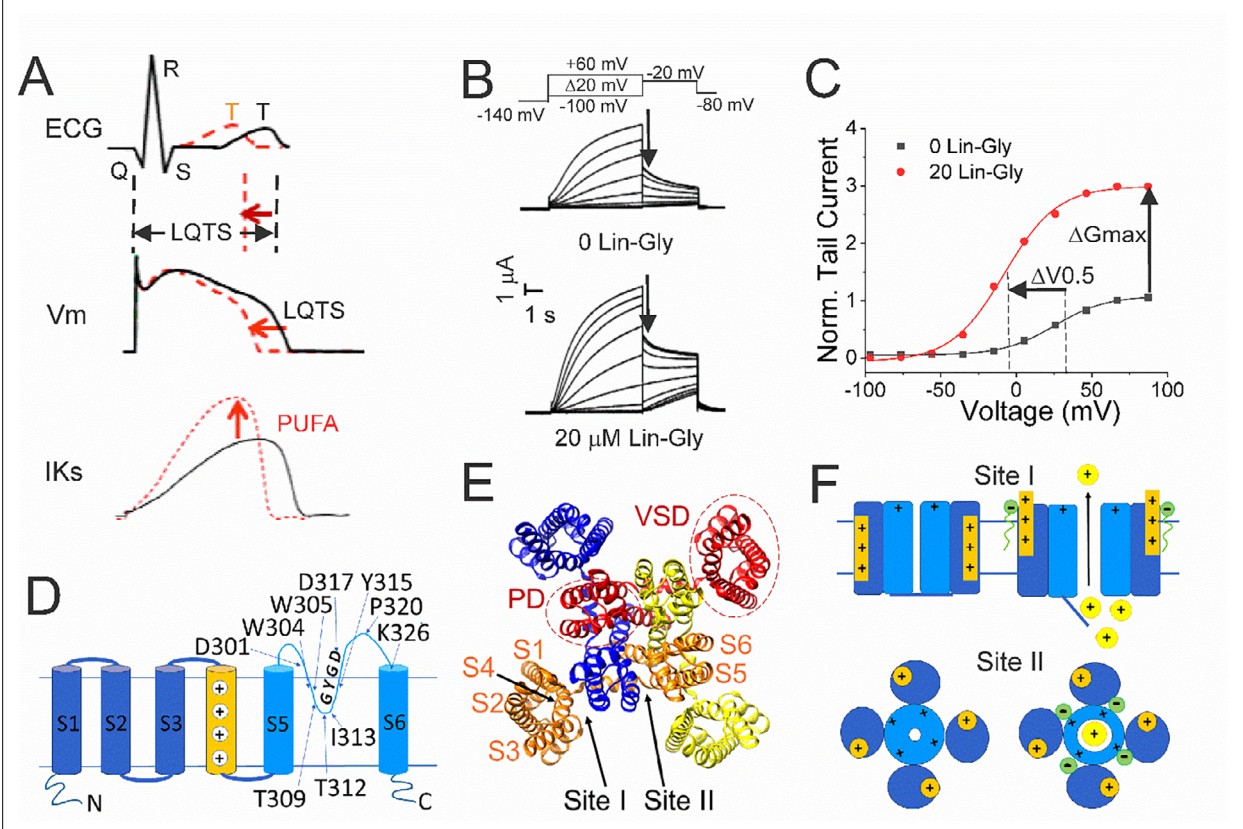

**Figure 1.** Polyunsaturated fatty acids (PUFAs) activate KCNQ1/KCNE1 channels. (**A**) (Black) Prolonged QT interval in the ECG is due to, for example, loss-of-function mutations of KCNQ1/KCNE1 channels that generate the IKs current that normally contributes to the repolarizing phase of the ventricular action potential (AP). (Red) PUFAs are potent activators of KCNQ1/KCNE1 channels that can restore the normal functioning of the channel and restore the AP duration and the QT interval. (**B**) Representative current traces of KCNQ1/KCNE1 in 0 µM and 20 µM of Lin-Glycine. Voltage protocol on top. (**C**) Conductance versus voltage curves from tail currents (measured at arrows in **B**). Channel activation by PUFA results in two main effects: a shift of the voltage dependence of activation (ΔV0.5) and an increase in the channel maximum conductance (ΔGmax). (**D**) KCNQ1 transmembrane topology. Residues mutated in this study are labeled. (**E**) KCNQ1 top view (PDB: 6UZZ) with PUFA binding sites: Site I, at the voltage sensor domain (VSD); and Site II, at the pore domain. The four subunits are shown in four different colors. (**F**) Cartoon of PUFA mechanism of action. Site I, top panel. Electrostatic interactions between PUFA head groups and positively charged residues in S4 facilitate channel activation by stabilizing the outward state of S4. Site II, bottom panel. PUFA interaction with residues in the pore domain facilitates the increase in the maximum channel conductance.

The online version of this article includes the following source data for figure 1:

**Source data 1.** Tail current data used to generate *Figure 1C*.

the positive charges in the S4 segment and causes the channel to open at more negative potentials (*Liin et al., 2015*). Site II is found in the pore domain, where electrostatic interactions between the PUFA head group and the positively charged residue, K326, facilitate an increase in the maximal conductance of the channel (*Liin et al., 2018*; *Figure 1E and F*).

Although previous studies have shown the importance of K326 for PUFA in increasing Gmax through binding at Site II (*Liin et al., 2018*), the mechanism by which PUFAs increase Gmax is still not understood. In this study, we use a combination of two-electrode voltage-clamp, single-channel recordings, site-directed mutagenesis, and recent cryoEM structures to gain insight into the molecular mechanism underlying the Gmax effect following PUFA binding at Site II. We propose a novel mechanism where the selectivity filter of KCNQ1 is normally unstable and the binding of PUFA to Site II stabilizes a network of interactions at the selectivity filter, which, in turn, leads to a more open and stable conformation of the KCNQ1 pore and thus an increase in channel open probability.

## Results

### Lin-Glycine drastically increases the chance of KCNQ1/KCNE1 channel opening

Lin-Glycine has been shown to increase the Gmax in whole-oocyte recordings of KCNQ1/KCNE1 channels 2.5-fold (*Bohannon et al., 2020*). To better understand how Lin-Glycine increases Gmax, we here extended our analysis to the single-channel level to study the behavior of KCNQ1/KCNE1 in the absence and presence of Lin-Glycine (20 µM) (*Figure 2*). Representative traces (*Figure 2A and B*) and all points histograms (*Figure 2C and D*) suggest that there is no change in single-channel conductance between KCNQ1/KCNE1 in control and with Lin-Glycine. This shows that the Gmax effect is not due to an increase in the single-channel conductance. However, Lin-Glycine caused a decrease in the latency to first opening (in control, 1.78 ± 0.36 s, n = 41 sweeps and after 20 µM Lin-Glycine, 1.03 ± 0.05 s, n = 99 sweeps; p=0.0025) and an increase in the number of non-empty sweeps (*Figure 2—figure supplement 1*). In control, there are a high number of empty sweeps as shown before for KCNQ1/KCNE1 channels (*Werry et al., 2013*). The average current during the single-channel sweeps was increased by 2.3 ± 0.33 times (n = 4 patches, p=0.006) by the application of Lin-Glycine (*Figure 2E*). The number of non-empty sweeps increased 2.85-fold (2.85 ± 0.85, n = 200 sweeps from four patches; p=0.001), going from a total of 41/200 non-empty sweeps in control to 99/200 when Lin-Glycine was applied. In contrast, once the channel opens, it seems to behave very similarly in control and Lin-Glycine because, in non-empty sweeps, the open probability (Po) in the last second of the traces was almost identical between control and Lin-Glycine conditions (Po = 0.78 ± 0.02 [n = 8 sweeps] in control vs 0.87 ± 0.04 [n = 8 sweeps] in Lin-Glycine) (*Figure 2F*). This high open probability was maintained in control conditions even for longer depolarizing voltage pulses, as if once the channels had opened they stayed open for the remaining of the voltage steps (*Figure 2—figure supplement 2*). The 2.85-fold increase in the number of non-empty sweeps is very similar to the Gmax increase seen in macroscopic currents.

The decrease in first latency is most likely due to an effect of Lin-Glycine on Site I in the VSD and related to the shift in voltage dependence caused by Lin-Glycine. In contrast, the increase in the number of non-empty sweeps is most likely an effect on Site II in the pore and related to the Gmax effect. We conclude that the Gmax effect of Lin-Glycine on KCNQ1/KCNE1 is mainly due to an increase in the Po by increasing the number of non-empty sweeps.

### Crevice residues affect PUFA ability to increase Gmax

It was previously shown that PUFAs binding at Site II electrostatically interact with the positively charged residue K326, located just outside the selectivity filter (*Liin et al., 2018*). We tested whether another residue very close to K326, the aspartic acid at position 301, is important for the PUFA interaction at Site II. Electrophysiological analysis revealed that when KCNQ1_WT/KCNE1 was mutated to KCNQ1_D301E/KCNE1, the ΔV0.5 was shifted similarly to the WT channel (*Figure 3—figure supplement 1*) but a dramatic decrease of the Gmax effect of Lin-Glycine was observed, showing the importance of this residue for PUFAs interaction at Site II (*Figure 3A–C*). However, it is not clear how PUFA binding at Site II increases Gmax.

In our previous MD simulations (*Yazdi et al., 2021*) based on the cryoEM structure of KCNQ1 with S4 in the activated state, PUFA binds to residues K326 and D301 at Site II that delimit a narrow crevice (*Figure 3D and E*), where PUFA could fit with both the head and the tail (*Figure 3D and E*). However, in the recent cryoEM structure of KCNQ1 with S4 in the resting state (*Mandala and MacKinnon, 2023*) the crevice between K326 and D301 is now so narrow that PUFA seems unable to fit into it (*Figure 3F*). In addition, there are large rearrangements in the selectivity filter when comparing the cryoEM structures with S4 activated (PDB: 8SIK) and S4 resting (PDB: 8SIN) (*Figure 4*). We have made a video showing the changes and the reorganization of the pore between these two structures (*Video 1*). The conformation of the pore significantly varies between what seems like a conductive (PDB: 8SIK) and a non-conductive (PDB: 8SIN) state of the channel. For example, in the structure with S4 in the resting state, Y315 and D317 have swung outward and left their positions in which they made hydrogen bonds with two tryptophans, W305 and W304, respectively. Y315 belongs to the K$^+$ channels signature sequence, a stretch of eight amino acids including TXXTXGYG, highly conserved among K$^+$ channels of different families and D317 is just next to this sequence (*Sansom et al., 2002; Heginbotham et al., 1994*). W304 and W305 belong to the aromatic ring cuff (*Figure 4C and D*) that

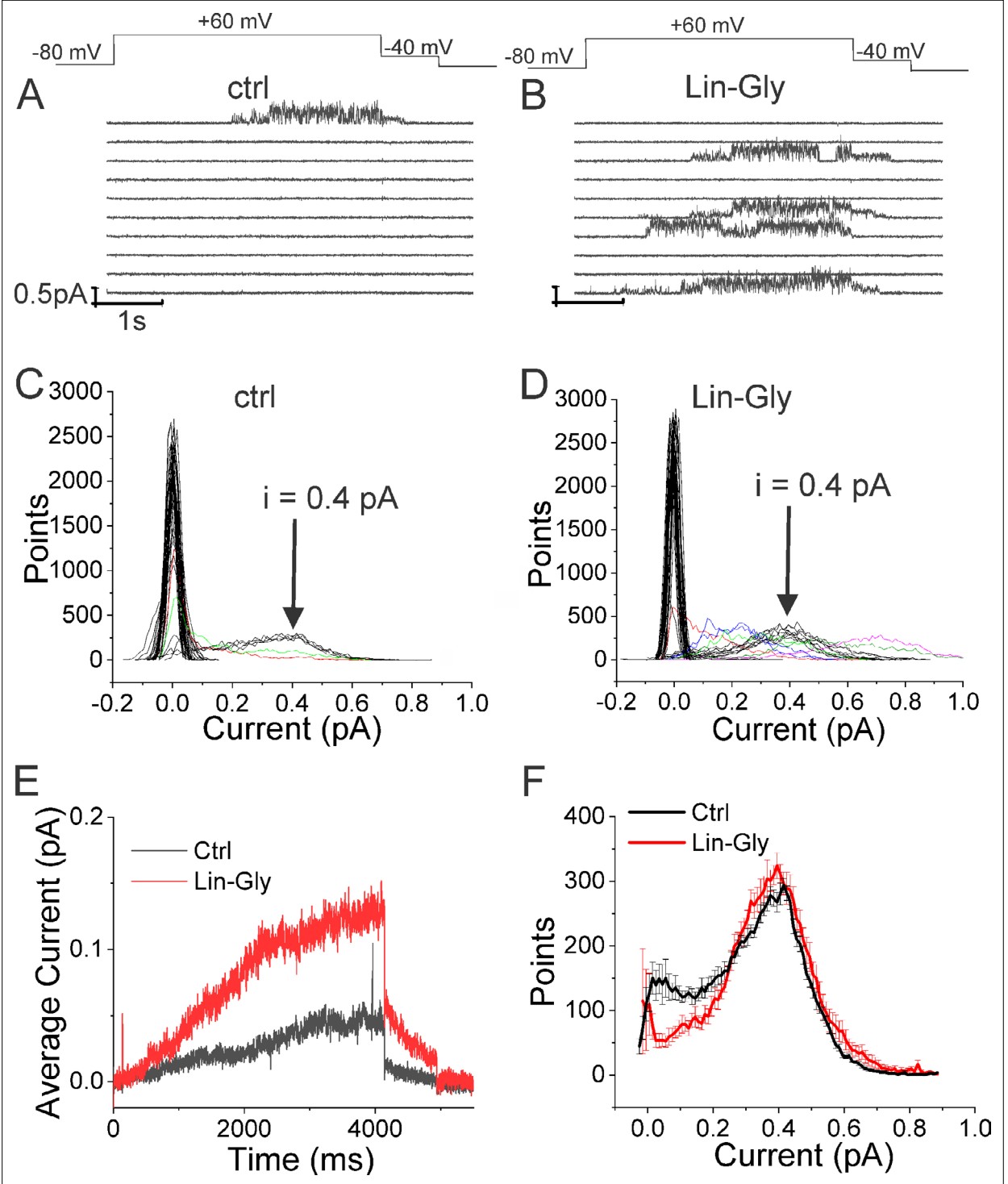

**Figure 2.** Lin-Glycine induces an increase in the Po of KCNQ1/KCNE1. (**A, B**) Ten consecutive traces of KCNQ1/KCNE1 in (**A**) control and (**B**) in the presence of Lin-Glycine (20 μM). (Top) Protocol used for the recordings. (**C, D**). All-point amplitude histogram of 50 consecutive traces in (**C**) control and (**D**) Lin-Glycine. Note no change in the single-channel current amplitude; however, an increase in the number of sweeps with channel opening is observed. Note that there were at least two channels in this patch. Different sweeps were assigned different colors to better visualize different types of channel behaviors. Note panels (**A–D**) are all from the same patch. (**E**) Average currents of 100 sweeps in control and Lin-Glycine. (**F**) All-point histogram of the last second of non-empty sweeps in control and in Lin-Glycine. We estimated the open probability from the all-point amplitude histogram by Po = Sum (iN/(i$_{estimate}$N$_{total}$)), where N is the number of points for a specific current i in the histogram, i$_{estimate}$ = 0.4 pA from the peak of the histogram, and

*Figure 2 continued on next page*

*Figure 2 continued*

$N_{total}$ = 10,000 is the total number of points in the last second of the trace. p=0.78 ± 0.02 (n = 8 sweeps) and p=0.87 ± 0.04 (n = 8 sweeps) for Control and Lin-Glycine, respectively, from the same patch.

The online version of this article includes the following figure supplement(s) for figure 2:

**Figure supplement 1.** First latency to opening is shortened by Lin-Glycine in wt KCNQ1/KCNE1 channels.

**Figure supplement 2.** Open probability stays high in KCNQ1/KCNE1 channels once opened.

has been suggested to be important for the stability of the open state of the selectivity filter in Kv channels (*Doyle et al., 1998*; *Larsson and Elinder, 2000*).

In addition, P320, which in the structure with an activated S4 sits in between W304 and W305 from two subunits (*Figure 4C*), has also swung outward in the structure with S4 in the resting state and exposed a gap between W304 and W305 in the aromatic cuff (*Figure 4D*, see also *Video 2*). The homologous proline in other Kv channels sits between the two tryptophans and stabilizes the aromatic ring cuff. *Figure 4—figure supplements 1 and 2* show the extent of the movement for each residue in the selectivity filter and nearby regions between the two conformations.

Based on our single-channel data with many empty sweeps, we propose the hypothesis that the KCNQ1 selectivity filter is inherently unstable and transitions between the two conformations – one conductive state that generates the non-empty sweeps and one non-conductive state that generates the empty sweeps in the single-channel data. We further propose that PUFA binding to residues at Site II stabilizes and promotes the conductive state of the pore. If PUFA binding to K326 and D301 promotes a series of interactions that stabilize the pore, then residues at the upper part of the selectivity filter, such as Y315 and D317, might be involved in those interactions that promote an open-conductive conformation of the channel.

## Residues that stabilize the selectivity filter are necessary for the Gmax effect

If our pore stability hypothesis is correct, we should see altered effects of Lin-Glycine on Gmax when residues important for the PUFA-promoted open-conductive conformation are mutated. We therefore made the mutations Y315F and D317E in KCNQ1. We tested Lin-Glycine on KCNQ1_Y315F/KCNE1 and KCNQ1_D317E/KCNE1 channels and compared the Gmax effect to that of the WT channel. The effect of Lin-Glycine was reduced on both mutant channels, with a particularly dramatic decrease for Y315F (*Figure 5A*). For KCNQ1_Y315F/KCNE1, we compared the effect on the voltage dependence of activation (ΔV0.5) and we observed a similar effect as for the WT channel (*Figure 5—figure supplement 1*), with both showing an ~–25 mV shift. We also tested the Y315F mutation at the single-channel level. As expected, Lin-Glycine did not increase the number of non-empty sweeps in KCNQ1_Y315F/KCNE1 channels (52/478 [10.9 % from three patches] of traces were non-empty in control and 44/533 [8.3% from three patches] of traces were non-empty in Lin-Gly) (*Figure 6*, *Figure 6—figure supplement 1*). The mutation Y315F reduced the single-channel current slightly from I = 0.4 pA in wildtype to I = 0.3 pA in the Y315F mutation (*Figure 6C and D*). The current average of all traces showed no increase in average current by Lin-Glycine (*Figure 6E*). This data shows that Y315 and D317 are necessary for the ability of Lin-Glycine to increase Gmax. Conductance vs voltage curves (G–V) for WT and mutant channels are shown in *Figure 5—figure supplement 2*.

We also found that another residue in the pore helix, T309, was important for the Gmax effect of Lin-Glycine. The homologous residue in the Shaker channel was earlier suggested to be important for the stabilization of the selectivity filter by hydrogen bonding to one of the tryptophans in the aromatic ring (*Pless et al., 2013*). We generated the mutant channel KCNQ1_T309S/KCNE1 and measured the effect of Lin-Glycine. As shown in *Figure 5*, the effect of Lin-Glycine on Gmax of the KCNQ1/KCNE1 mutant channel was noticeably reduced compared to the WT channel showing that this residue contributes to the Gmax effect (*Figure 5A*). We also tested the involvement of proline, which makes up a part of the aromatic cuff in the activated state of the channel (*Figure 4C*) by creating the mutant channel KCNQ1_P320L/KCNE1 and found a significant reduction of the Gmax effect for this mutant (*Figure 5A*). All these data suggest that mutations of residues at the outer portion of the selectivity filter do affect the Gmax increase by Lin-Glycine. These data are consistent with our hypothesis that PUFAs increase the Gmax by affecting interactions that stabilize the selectivity filter.

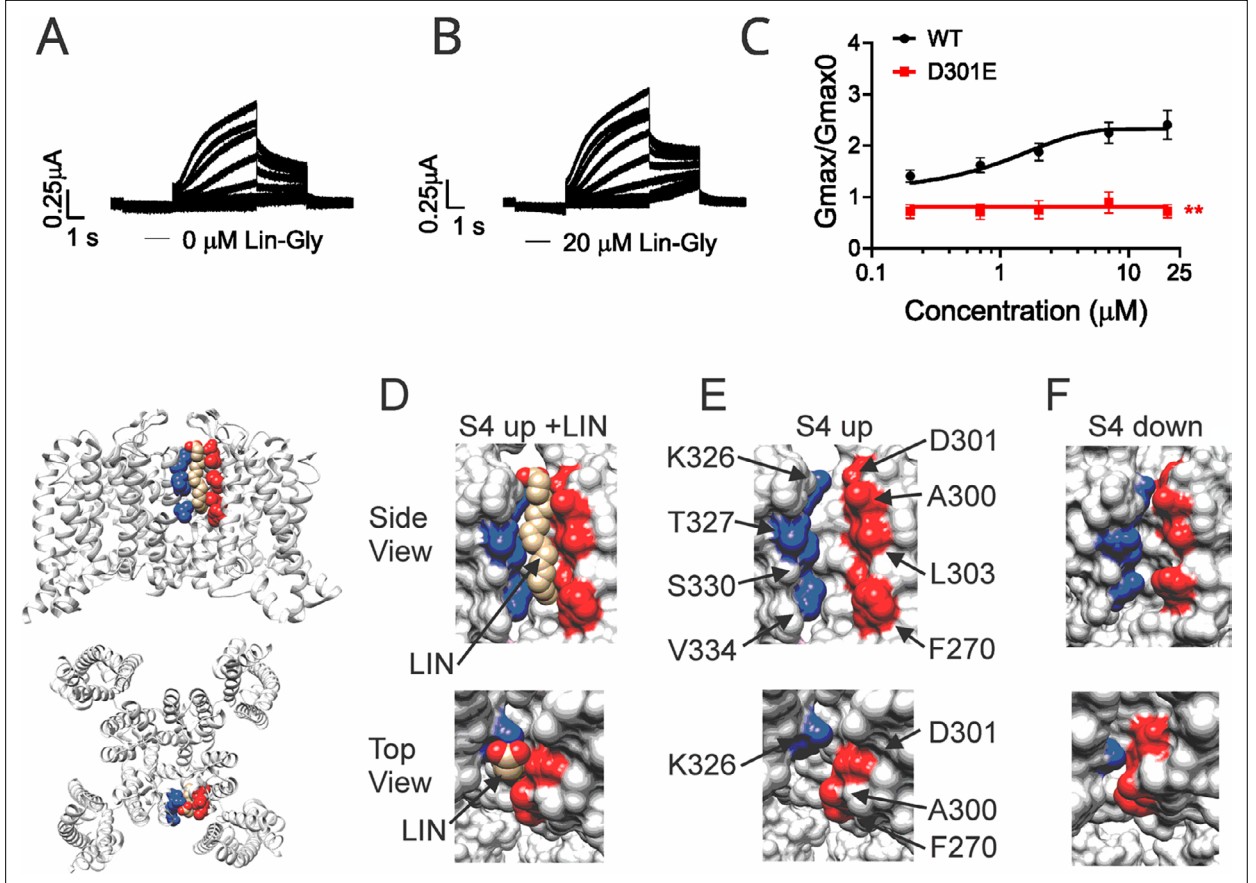

**Figure 3.** Polyunsaturated fatty acid (PUFA) binds to a state-dependent small crevice between K326 and D301. (**A, B**) KCNQ1_D301E/KCNE1 representative current traces (**A**) in 0 μM of Lin-Glycine. (**B**) After perfusion of 20 μM of Lin-Glycine. (**C**) Gmax/Gmax0 for KCNQ1/KCNE1 channels and KCNQ1_D301E /KCNE1 channels. Gmax/Gmax0 was significantly reduced for the D301E mutation compared to WT channels (p=0.0018, n = 4 oocytes). Student's *t*-test was used to conduct statistical analysis. (**D–F**) Structures of crevice between S5 and S6 in KCNQ1 with S4 activated (S4 up) (**D, E**) and S4 resting (S4 down) (**F**). Residues that surround the crevice from S6 shown in blue (K326, T327, S330, V334) and from the pore helix and S5 in red (D301, A300, L303, F270). Remaining KCNQ1 residues shown in light gray. On the left is shown the location of the crevice in KCNQ1 (top) side view and (bottom) top view. (**D**) In MD simulations (*Yazdi et al., 2021*), linoleic acid (LIN: gold color) fits in a narrow crevice present in the cryoEM structure of activated state KCNQ1. (**E**) Same view as in (**D**) but without LIN. (**F**) In the cryoEM structure with S4 in the resting state, the crevice between K326 and D301 is too narrow to fit LIN.

The online version of this article includes the following source data and figure supplement(s) for figure 3:

**Source data 1.** Gmax data used to generate *Figure 3C*.

**Figure supplement 1.** Effect of Lin-Glycine in shifting the voltage dependence of activation (ΔV0.5).

**Figure supplement 1—source data 1.** ΔV0.5 data used to generate the figure.

The residues involved in the Gmax effect are found near the external region of the selectivity filter, suggesting that the network of interactions that are altered during the switch from non-conductive to conductive state is confined to the outer region of the selectivity filter and pore helix of KCNQ1/KCNE1 channels. To test the specificity of the network localization, we mutated two residues, T312 and I313, in the internal portion of the selectivity filter. As a confirmation of our hypothesis, we found that for both KCNQ1/KCNE1 mutant channels, T312S and I313S, the effect of Lin-Glycine in increasing Gmax resembled values obtained in the WT channels (2.49 ± 0.98 and 2.18 ± 0.05, respectively) (*Figure 5B*). Mutations of residues in the more intracellular region of the selectivity filter do not affect the Gmax increases, as if the interactions that stabilize the channel involve residues located near the external region of the selectivity filter.

Taken together, our results suggest that the binding of PUFA to Site II increases Gmax by promoting a series of interactions that stabilize the channel pore in the conductive state. For instance, we speculate that in the conductive state, hydrogen bonds between W304-D317 and W305-Y315, which are

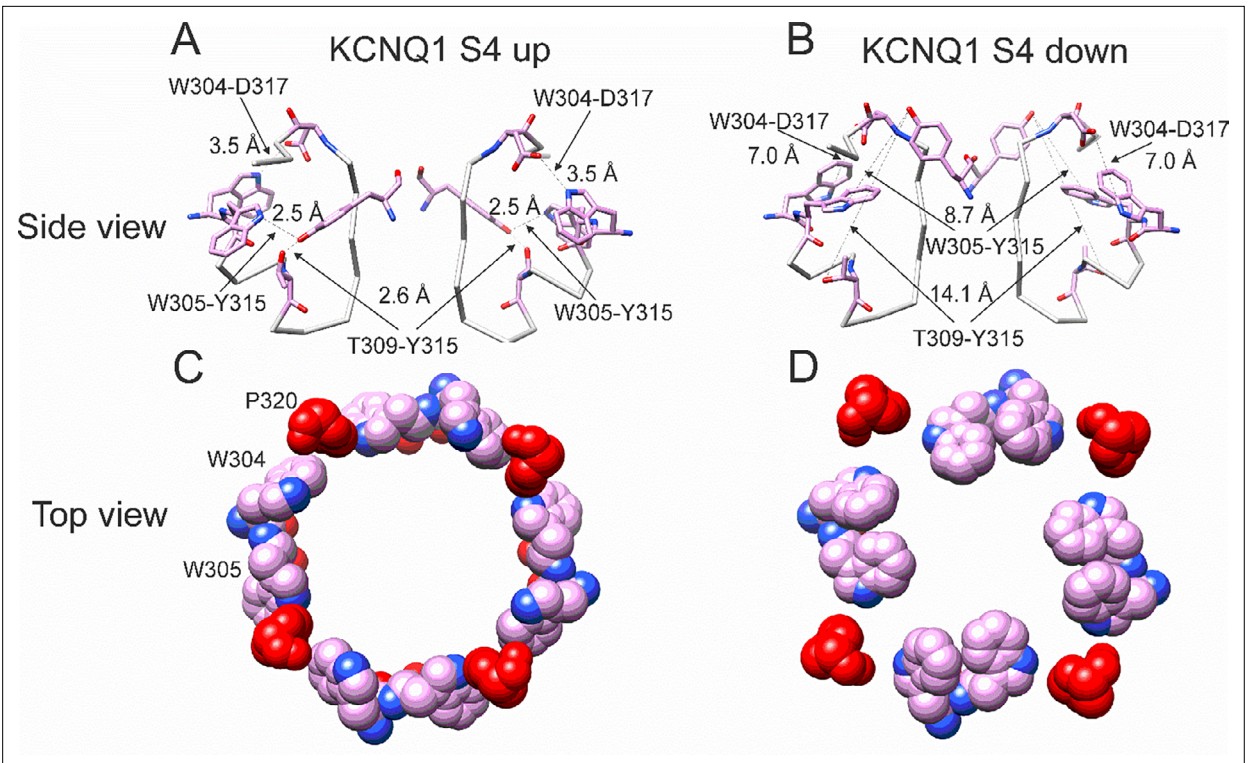

**Figure 4.** Different conformations of selectivity filter in cryoEM structures with S4 activated or resting. (**A**) Selectivity filter of KCNQ1 with S4 activated (S4 up; PDB: 8SIK). Distances between D317 and W304 and between T309 and Y315 are short enough to form hydrogen bonds (dashed lines). (**B**) Selectivity filter of KCNQ1 with S4 in resting state (S4 down; PDB: 8SIN). Distances between D317 and W304 and between T309 and Y315 are too long for hydrogen bonds (dashed lines). Only two subunits are shown for clarity. (**C, D**). Aromatic cuff in KCNQ1 with (**C**) S4 activated and (**D**) resting S4. Note how P320 (red) moves away from its position in between W304 and W305 from two different subunits in the S4 down conformation.

The online version of this article includes the following figure supplement(s) for figure 4:

**Figure supplement 1.** Atomic displacements at the selectivity filter and nearby regions in the transition from up to down state.

**Figure supplement 2.** Graphic indicating the atomic displacement of residues at the selectivity filter and nearby regions in the transition from up to down state.

**Figure supplement 3.** Comparison between the aromatic ring cuff configuration of Shaker and KCNQ1 channels.

likely absent in the non-conductive conformation of KCNQ1, are created and that PUFA binding to K326 and D301 at Site II favors the transition toward the conductive state of the channel (*Figure 7*).

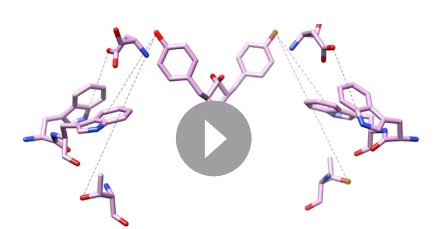

**Video 1.** Large conformational changes in selectivity filter between Kv7.1 structures with S4 up and S4 down. Interpolation between the cryoEM structures of Kv7.1 with S4 in the activated conformation (PDB 8SIK) and S4 in the resting conformation (PDB SIN) showing the position of residues W304, W305, T309, Y315, and D317 that are important for the stability of the selectivity filter in related Kv channels. Dashed lines between residues proposed to make hydrogen bonds in Kv channels.
https://elifesciences.org/articles/95852/figures#video1

## Discussion

We have previously proposed models in which the effect of PUFAs on IKs channels involve the binding of PUFAs to two independent sites: one at the voltage sensor (Site I) and one at the pore domain (Site II). The result is a potent activation of IKs channels, with an increase in the maximum conductance by binding to Site II and a shift of the voltage dependence of activation of the channel by binding to Site I. The mechanism of the voltage shift effects at Site I, where the PUFA head group electrostatically interacts with positively charged S4 residues and thereby facilitates

channel activation, has been investigated in previous studies (*Yazdi et al., 2021*; *Bohannon et al., 2023*; *Liin et al., 2018*; *Jowais et al., 2023*). However, less is known about the molecular mechanism by which PUFA increases the maximum conductance (Gmax) of the channel following binding to Site II. A positively charged lysine at position 326 was suggested to be critical for PUFA-channel interaction, as mutations of K326 completely abolished the Gmax effect (*Liin et al., 2018*). We have here shown that the interaction at Site II also involves another residue, the aspartic acid 301, as mutagenesis analysis revealed that the effect of Lin-Glycine on Gmax was abolished when the residue was substituted with a glutamic acid.

Our single-channel recordings revealed that Lin-Glycine decreases the latency to first opening and increases the Po. The decrease in the latency to first opening is most likely due to the binding of Lin-Glycine to Site I at the VSD and related to the shift in voltage dependence caused by Lin-Glycine (*Figure 2—figure supplement 1*). In addition, our single-channel recordings revealed that Lin-Glycine did not change the single-channel conductance, but instead increased Gmax by increasing the Po of the channel. This increase in Po was mainly due to an increase in non-empty sweeps when Lin-Glycine was applied in comparison to control solutions.

We tested the hypothesis that the effect of Lin-Glycine involved conformational changes in the selectivity filter following PUFA binding to two residues K326 and D301 at the pore domain. Those residues delimit a small crevice that seems to change in size in different structures with S4 activated or resting (*Figure 3D–F*) and seem to affect PUFA's ability to increase Gmax. We made several mutations in residues known to stabilize the selectivity filter of potassium channels (Y315F, D317E, T309S, and P320L; *Figure 5A*). The Lin-Glycine effect on the Gmax was much reduced in these mutants, suggesting that these residues are necessary for the Gmax effect. To gain further insight into the molecular interactions that could underline the Gmax effect by PUFA binding to Site II, we used the latest KCNQ1 structure with the S4 segment in the resting state (*Mandala and MacKinnon, 2023*). The selectivity filter in this structure showed a very different conformation of the pore region and selectivity filter of KCNQ1 compared to the structures with activated VSDs. Clearly, there will be other differences in the pore domain between structures with activated and resting VSDs, for example, the state of the activation gate. Many of the interactions that have been shown to be important for the stability of the selectivity filter are missing in the KCNQ1 structure with a resting VSD. These changes in the selectivity filter can best be seen in our interpolation video between the states with the S4 segments moving from the resting state to the activated state (*Video 1*). This gave us the idea that the effect of PUFAs in increasing the maximum conductance of the KCNQ1 channel is linked to their ability to stabilize the pore of the channel in a conductive state. It was previously shown that several interactions at the pore region of K⁺ channels are important for ensuring channel conductivity. For example, a feature conserved among K⁺ channels is the aromatic ring cuff that stabilizes the conducting state and plays a role in C-type inactivation (*Doyle et al., 1998*; *Larsson and Elinder, 2000*; *Pless et al., 2013*; *Kurata and Fedida, 2006*). This structure is made up of two tryptophan residues (W) and a proline residue (P) that sits in between the two tryptophan residues (*Video 2*). In Shaker and KcsA K⁺ channels, the two tryptophans are involved in C-type inactivation and modification of those interactions manipulate the extent of inactivation (*Cordero-Morales et al., 2011*; *Yang et al., 1997*). In Shaker, breaking the hydrogen bonds between the two tryptophans of the aromatic ring cuff and the aspartic acid and tyrosine at the selectivity filter causes an acceleration of the rate of C-type inactivation (*Pless et al., 2013*). In KCNQ1, the two tryptophans of the aromatic ring cuff correspond to W304 and W305. Another conserved residue in K⁺ channels important for the stability of the selectivity filter is P320 (equivalent to P450 in Shaker) that makes up

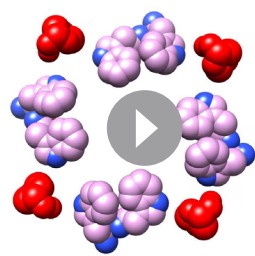

**Video 2.** Large conformational changes in aromatic cuff between Kv7.1 structures with S4 up and S4 down. Interpolation between the cryoEM structures of Kv7.1 with S4 in the activated conformation (PDB 8SIK) and S4 in the resting conformation (PDB SIN) showing the position of the aromatic cuff residues W304, W305, and P320 (red) that are important for the stability of the selectivity filter in related Kv channels.
https://elifesciences.org/articles/95852/figures#video2

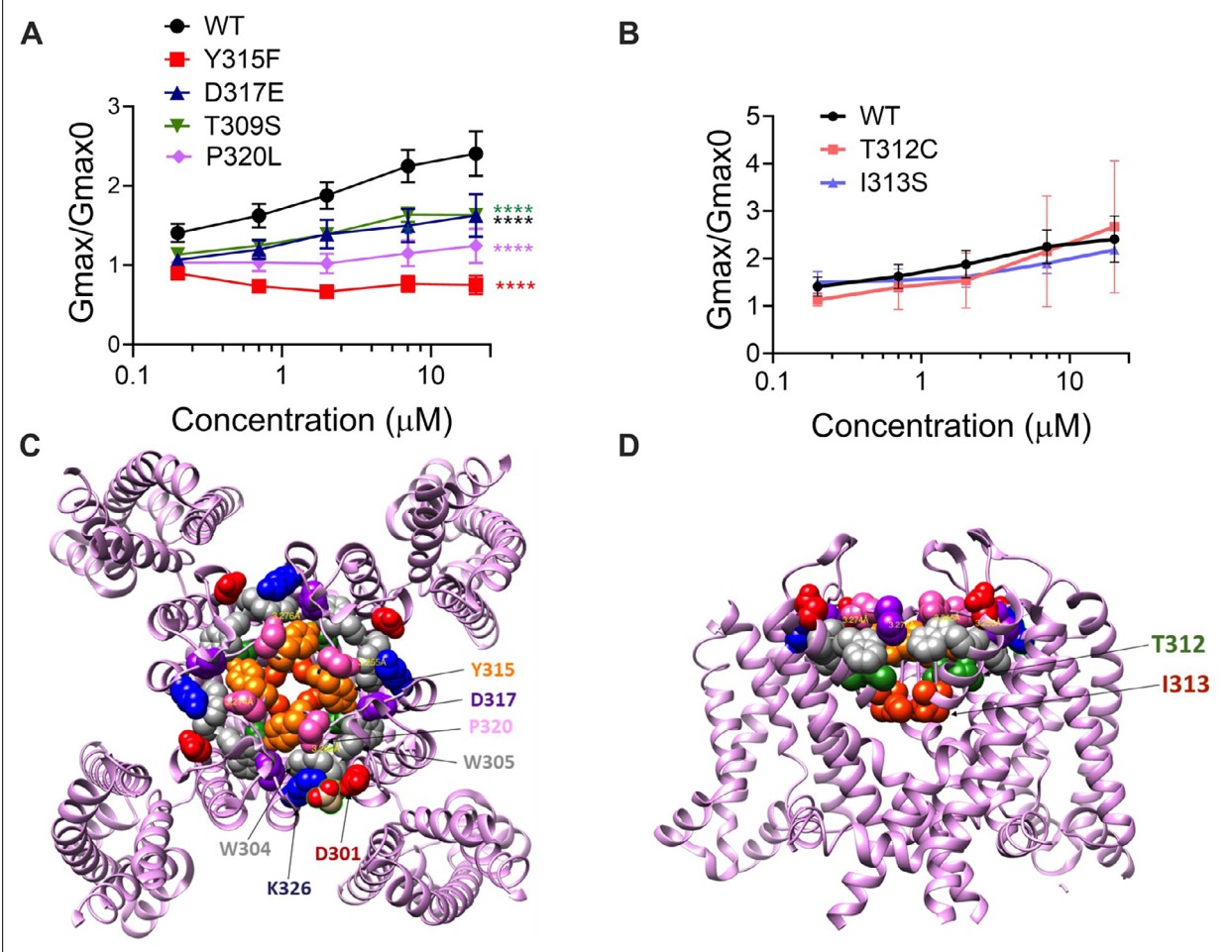

**Figure 5.** The ability of Lin-Glycine to increase the channel conductance is reduced when channel pore residues are mutated. (**A**) Gmax/Gmax0 values obtained for KCNQ1_WT/KCNE1 channel (black) and mutant channels. KCNQ1_Y315F/KCNE1 (red), KCNQ1_D317D/KCNE1 (blue), KCNQ1_T309S/KCNE1 (green), and KCNQ1_P320L/KCNE1 (purple), the Gmax/Gmax0 is significantly reduced (p<0.0001 n = 4 oocytes). One-way ANOVA with Dunnett's post hoc multiple-comparisons test was used for statistical analysis. (**B**) Gmax/Gmax0 values for KCNQ1_T312C/KCNE1 and KCNQ1_I313S/KCNE1 (p=0.14 and p=0.10, respectively) (n = 3 oocytes). One-way ANOVA with Dunnett's post hoc multiple-comparisons test was used for statistical analysis. (**C, D**) Top view and side view of KCNQ1 channel with mutated residues highlighted.

The online version of this article includes the following source data and figure supplement(s) for figure 5:

**Source data 1.** Gmax data used to generate *Figure 5A and B*.

**Figure supplement 1.** Effect of Lin-Glycine in shifting the voltage dependence of activation (ΔV0.5).

**Figure supplement 1—source data 1.** ΔV0.5 data used to generate *Figure 5—figure supplement 1*.

**Figure supplement 2.** GV curves for KCNQ1 mutants.

**Figure supplement 2—source data 1.** Gmax data used to generate *Figure 5—figure supplement 2A–G*.

part of the aromatic ring cuff. In the transition between the non-conductive and conductive state of KCNQ1, the proline pulls away from its position close to W304 and W305 and flips outward (*Video 2*). We hypothesize that those hydrogen bonds between W304-D317 and W305-Y315 and the proline interactions are likely absent in the non-conductive conformation of KCNQ1, thereby generating the high number of empty sweeps in control conditions. Occasionally the channel will reform these bonds and interactions and transition into the conductive conformation, thereby generating the non-empty sweeps. We hypothesize that the binding of Lin-Glycine to K326 and D301 at Site II biases the transition toward the conductive conformation, thereby increasing the number of non-empty sweeps and increasing Gmax. We here show that mutations of residues involved in these hydrogen bonds in the

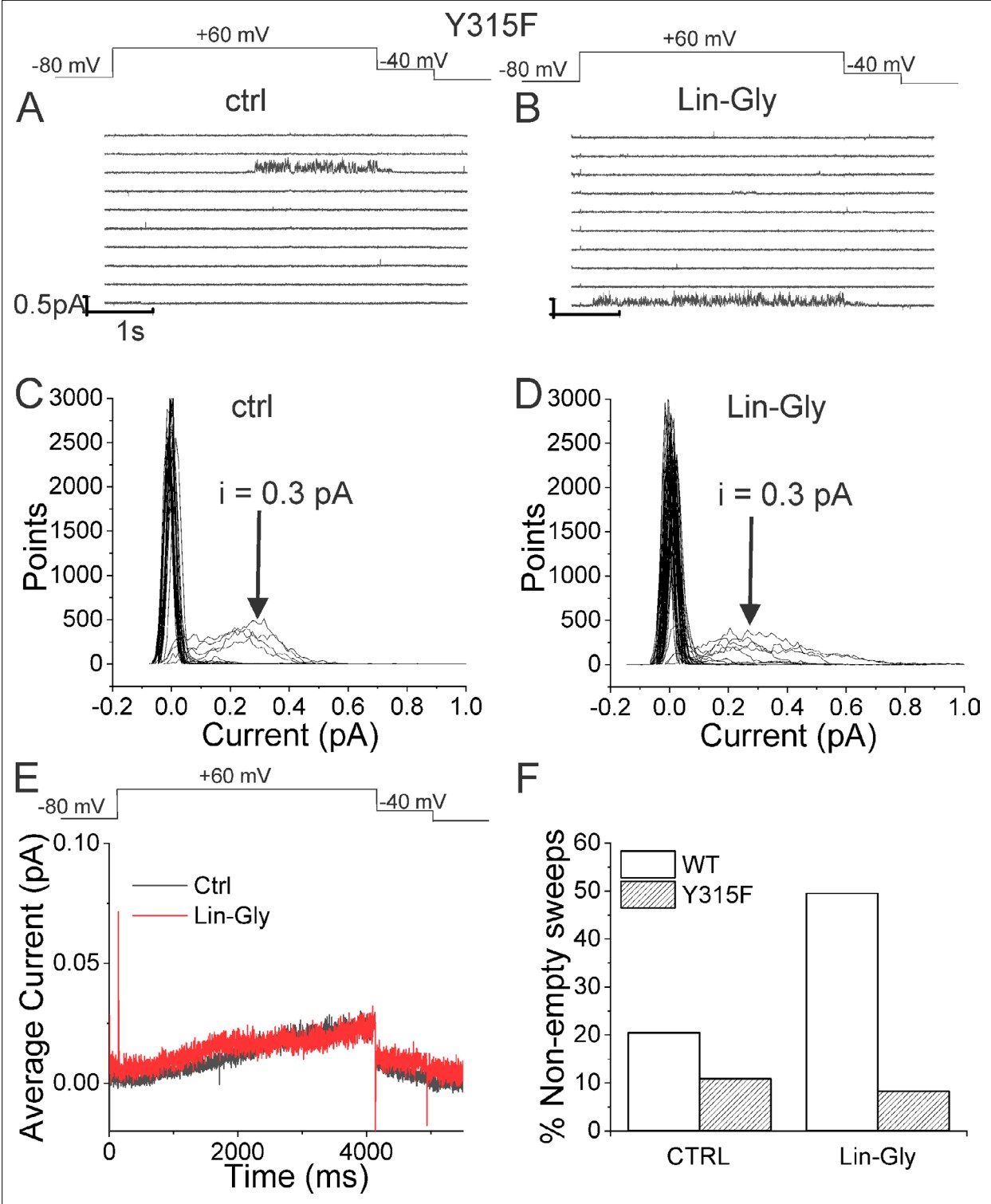

**Figure 6.** Lin-Glycine does not increase the Po of KCNQ1_Y315F/KCNE1. (**A, B**) Ten consecutive traces of KCNQ1_Y315F/KCNE1 in (**A**) control and (**B**) in the presence of 20 µM Lin-Glycine. (Top) Protocol used for the recordings. (**C, D**) All-point amplitude histogram of 50 consecutive traces in (**C**) control and (**D**) Lin-Glycine. The single-channel current amplitude was reduced to 0.3 pA compared to 0.4 pA for WT KCNQ1/KCNE1 (*Figure 2C and D*). Note that there were at least two channels in this patch. (**E**) Average currents of 478 sweeps in control and 533 sweeps in Lin-Glycine. Note that panels (**A–D**) were all from the same patch.

The online version of this article includes the following source data and figure supplement(s) for figure 6:

**Source data 1.** Percentage of non-empty sweeps used to generate *Figure 6F*.

*Figure 6 continued on next page*

*Figure 6 continued*

**Figure supplement 1.** First latency to opening is not shortened by Lin-Glycine in Y315Y channels.

selectivity filter greatly reduce or abolish the Gmax effect of Lin-Glycine, as if Lin-Glycine increases the Gmax by stabilizing a conductive state of the selectivity filter through these hydrogen bonds.

We noticed that the arrangement of the aromatic ring cuff is slightly different between Shaker K⁺ channels and KCNQ1 channel. For instance, in Shaker, the proline residue sits in between the two tryptophan residues and stabilizes the aromatic ring cuff (*Figure 4—figure supplement 3A*). In contrast, in KCNQ1 the proline is positioned a little further outward and away from the two tryptophan residues, thus generating a looser arrangement of the aromatic ring cuff (*Figure 4—figure supplement 3B*). This difference might contribute to rendering the pore of KCNQ1 more unstable, resulting in a large number of empty sweeps and the characteristic flickering nature of channel openings (*Eldstrom et al., 2021*). In Shaker channels, the aromatic cuff seems more stable since it displays few empty sweeps and less flicker during bursts. This difference in the stability of the aromatic cuff between Shaker and KCNQ1 might explain why the effect of PUFA on Gmax is large for KCNQ1 but not seen in Shaker (*Börjesson et al., 2008*; *Börjesson and Elinder, 2011*) even if the residues involved in the Gmax effect are conserved between these two channels.

Our single-channel data show that the KCNQ1/KCNE1 switches slowly (>10 s) between conductive and non-conductive states, giving rise to many empty sweeps and few active sweeps. However, once the channel becomes conductive during a depolarization, the channel stays conductive for the remainder of the sweep (see, e.g., *Figure 2—figure supplement 2* for 20 s sweeps), as if VSD activation stabilizes the conductive state. Therefore, we propose a model in which the selectivity filter is stabilized more in the non-conductive state when VSD is resting but stabilized more in the conductive

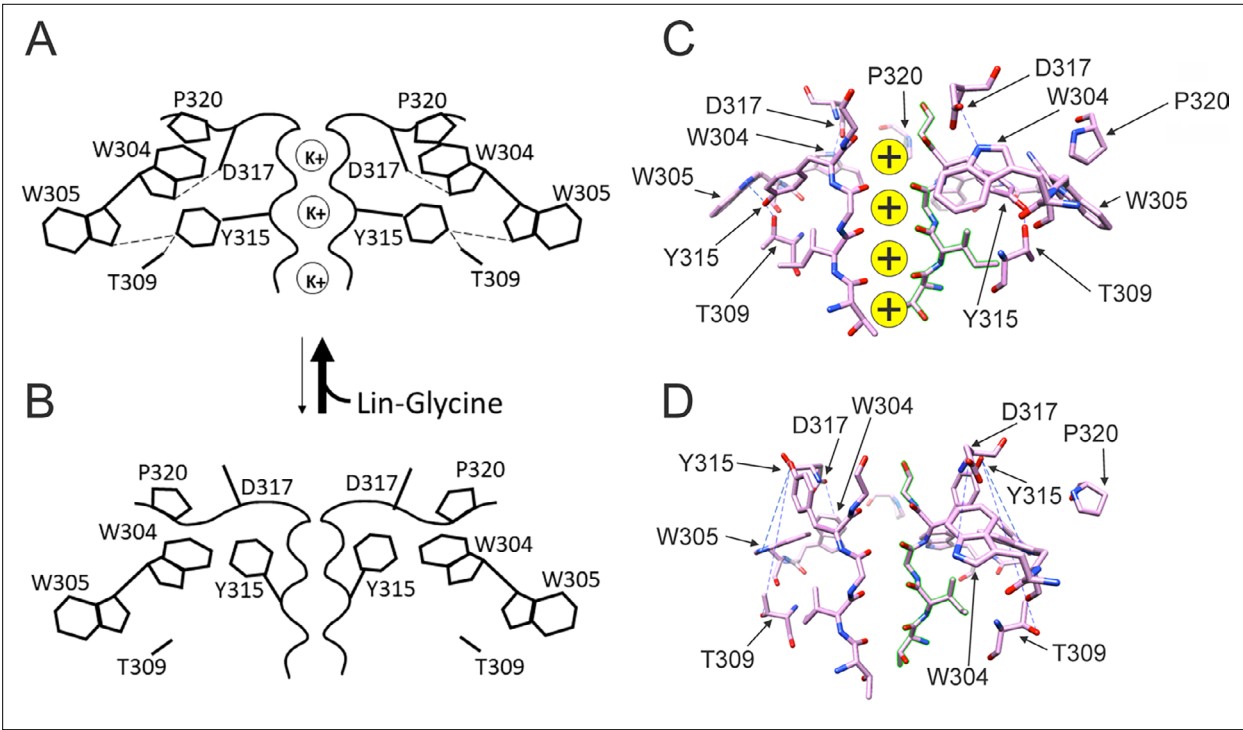

**Figure 7.** Conformational changes occurring at the pore during the transitions between non-conductive and conductive states. (**A**) The binding of polyunsaturated fatty acid (PUFA) to Site II between K326 and D301 induces a series of interactions between residues near the external part of the selectivity filter. W304-D317 and W305-Y315 form hydrogen bonds (dash line, black), Y315 interacts also with T309 (dash line, black), and P320 is reoriented to sit on top of W304 and W305 to favor a more stable configuration of the aromatic ring cuff. The result of those new interactions is a more stable and conductive pore. (**B**) In the non-conductive state, those interactions are likely to be absent and this results in a more unstable selectivity filter. Also, P320 is now flipped from its position on top of the two tryptophans of the aromatic ring cuff. (**C**) CryoEM selectivity filter with S4 in the activated-state representative of a conductive selectivity filter. (**D**) CryoEM selectivity filter with S4 in the resting state, representative of a non-conductive selectivity filter.

state when VSD is activated. This model is consistent with the cryoEM structures of KCNQ1 with VSD resting displaying a non-conductive selectivity filter and with VSD activated displaying a conductive selectivity filter (*Mandala and MacKinnon, 2023*). The KCNQ1 structure with VSD resting was obtained in a low K⁺ solution, which in other Kv channels would promote C-type inactivation of the selectivity filter. However, we think it is the resting state of the VSD, and not the low K⁺, that caused the non-conductive selectivity filter in the KCNQ1 structure because a KCNQ1 structure with VSD activated in the low K⁺ solution had a selectivity filter that was nearly identical to the KCNQ1 structure in high K⁺ with VSD activated (except for fewer K⁺ ions in the filter in low K⁺ solutions) (see Suppl. Figs. S5C and S5D in *Mandala and MacKinnon, 2023*). Note also the currents from KCNQ1/KCNE1 channels display little external K⁺ dependence (a small inhibition in addition to what is expected from changes in driving force) and that KCNQ1 channels are more strongly inhibited by high extracellular K⁺ concentrations (*Abrahamyan et al., 2023*; *Larsen et al., 2011*) in contrast to other Kv channels, such as Shaker K⁺ channels, which are inactivated more in low K⁺ (*Larsson and Elinder, 2000*; *Kurata and Fedida, 2006*). Our studies suggest that the mechanism of PUFAs to increase KCNQ1/KCNE1 maximum conductance relies on the ability of PUFAs to favor the conductive conformation of the selectivity filter: PUFAs promote interactions between residues in the selectivity filter that stabilize the channel pore. Furthermore, we identify a crevice between K326 and D301 that is present in structures of KCNQ1 with activated VSD but not in structures where the VSD is in the resting state. We propose that PUFA binding in this crevice stabilizes the network of interactions that form the conductive form of the selectivity filter. The result is a more stable and conductive pore, which explains the increase in Gmax by PUFAs.

## Materials and methods
### Molecular biology
KCNQ1 (UniProt: P51787) and KCNE1 (UniProt: P15382) cRNA were transcribed using the mMessage mMachine T7 kit (Ambion). KCNQ1 was co-expressed with KCNE1 subunit, following a 3:1, weight:-weight (Q1:E1) cRNA ratio to make up the KCNQ1/KCNE1 currents. Site-directed mutagenesis was performed using the QuikChange II XL Mutagenesis Kit (QIAGEN Sciences) for mutations in KCNQ1. Primer sequences for mutagenesis are (mutated residues are shown in bold and altered bases are in capital letters):

> D301E gagttcggcagctacgca**gaA**gcActgtggtgggggtgg,
> T309S ctgtggtggggggtCgtc**Tca**gtcacAaccatcggctatg,
> Y315F gtcaccaccatcggc**tTt**ggAgacaaggtgccc,
> D317E caccatcggctaCggA**gaG**aaggtgccccagacg,
> T312C gtggtcacagtcacc**tgc**atcggctatggggac,
> I313S gtcacagtcaccacc**TCc**ggctatggggacaag, and
> P320L ctatggggacaaggtg**cTc**cagacgtgggtcggg

50 ng of complementary RNA was injected into defolliculated *Xenopus laevis* oocytes (Ecocyte Bioscience, Austin; Xenopus 1 Corp, Dexter) for channel expression. After RNA injection, cells were incubated for 24–96 hr in standard ND96 solution (96 mM NaCl, 2 mM KCl, 1 mM MgCl₂, 1.8 mM CaCl₂, 5 mM HEPES), 1 M sodium pyruvate and penicillin (10,000 units)-streptomycin (10 mg/ml in 0.9% NaCl; pH 7.5) at 16°C before experiments.

### Two-electrode voltage clamp
KCNQ1/KCNE1 currents were recorded using the two-electrode voltage clamp technique. The recording chamber was filled with ND96 solution (96 mM NaCl, 2 mM KCl, 1 mM MgCl₂, 1.8 mM CaCl₂, and 5 mM HEPES; pH 7.5). Pipettes were filled with 3 M KCl. PUFA Lin-Glycine was purchased from Cayman Chemicals (Ann Arbor, MI), kept in stock of 100 mM with ethanol at –20°C and diluted in ND96 solution the day of the experiments. Electrophysiological recordings were acquired using Clampex 10.7 software (Axon, pClamp, Molecular Devices). Lin-Glycine was perfused into the recording chamber in a pre-application step until reaching steady state, followed by an (I–V) protocol to measure the current–voltage relationship before and after perfusion. During PUFA application, cells were held at –80 mV and depolarization from –80 mV to 0 mV (5 s) was applied every 30 s, before

stepping to –40 mV and then back to – 80 mV. In the I–V protocol, a hyperpolarized step from –80 mV to –140 mV is applied before 20 mV voltage steps from –100 mV to +60 mV. Tail currents are recorded at –20 mV before returning to holding potential.

### Single channel recordings

Single-channel currents were recorded from transiently transfected mouse *ltk*- fibroblast cells (LM cells) using 1.5 µl Lipofectamine 2000 (Thermo Fisher Scientific). LM cells were bought from ATCC (Manassas, VA: ATTC number CCL-1.3; lot number 63734321) who authenticated them (STR profiling) and confirmed that the cell lines tested negative for mycoplasma contamination. Cells were transfected with 1.5 µg of pcDNA3 containing a linked KCNE1-KCNQ1 construct (*Murray et al., 2016*) to ensure fully KCNE1-saturated complexes, in addition to a plasmid containing green fluorescent protein to identify transfected cells. Cells were recorded from 24 to 48 hr after transfection using an Axopatch 200B amplifier, a Digidata 1440A and pClamp10 software (Molecular Devices, San Jose, CA). The bath solution contained (in mM) 135 KCl, 1 $MgCl_2$, 1 $CaCl_2$, 10 HEPES, 10 dextrose (pH 7.4 with KOH). The pipette solution contained (in mM) 6 NaCl, 129 MES, 1 $MgCl_2$, 5 KCl, 1 $CaCl_2$, 10 HEPES (pH 7.4 with NaOH). Pipettes were pulled from thick-walled borosilicate glass (Sutter Instruments, Novato, CA) using a linear multistage electrode puller (Sutter Instruments), fire polished and coated with Sylgard (Dow Corning, Midland, MI). Electrode resistance was between 40 and 80 MΩ after polishing. Currents were sampled at 10 kHz, low-pass filtered at 2 kHz at acquisition and subsequently digitally filtered at 200 Hz for presentation and analysis. Data were collected using cell-attached patch configuration to minimize disruption to the patch and avoid rundown problems due to the loss of $PIP_2$. Lin-Glycine was solubilized in DMSO and added directly to the bath. Only patches that were largely free of endogenous currents and had few channels, such that there were several blank sweeps to average for use for leak subtraction, were analyzed.

### Data analysis

To measure the conductance vs voltage (G–V) curve, KCNQ1/KCNE1 tail currents were measured at –20 mV and obtained values were plotted against the activation voltages and fitted to a Boltzmann function:

$$G(V) = Gmin + (Gmax − Gmin)/\{1 + \exp[−(V − V_{50})]/s\}$$

where $G_{min}$ is the minimal conductance, $G_{max}$ is the maximal conductance, $V_{50}$ the midpoint (which describes the voltage at which the conductance is half the maximal conductance established from the fit), and *s* is the slope of the curve in mV. The difference in Gmax effect, before and after application of Lin-Glycine in each oocyte, is used as a measure of the change in maximal conductance. To understand the concentration dependence of LIN-Glycine effect on $G_{max}$, the following concentration–response curve was fitted to the data:

$$G_{max}/G_{max}^0 = 1 + B/[1 + (PUFA_{50}/[PUFA])^H]$$

where B is the maximum relative increase in $G_{max}$ {($G_{max}$ - $G_{max0}$)/$G_{max0}$}, $PUFA_{50}$ the PUFA concentration needed to cause 50% of the maximal effect, and H is the Hill coefficient. Average values are expressed as mean ± SEM and n represents the number of experiments (unless specified).

Statistical analysis was conducted using GraphPad Prism 8 (GraphPad Software, Boston, MA). Statistical tests used were one-way ANOVA with Dunnett's post hoc multiple-comparisons test (*Figure 5*) and Student's *t*-test for all other single comparisons. Number of recordings, n, always represents biological replicates. Exclusion criteria were pre-established. Cells were excluded if not expressing enough channels (<0.5 µA at +60 mV) or if not well voltage clamped (>5 mV errors).

Data were analyzed using Clampfit 10.7 (pCLAMP), Origin Pro (OriginLab Corporation), and GraphPad Prism 8 software (GraphPad Software).

### Material availability

Materials (e.g., DNA of Kv7.1 mutants) are available under reasonable requests from scientists. Data is available as Data Source files.

## Additional information

### Funding

| Funder | Grant reference number | Author |
| --- | --- | --- |
| National Heart, Lung, and Blood Institute | 5R01HL131461 | H Peter Larsson |
| Heart and Stroke Foundation of Canada | G-21-0031566 | David Fedida |
| Canadian Institutes of Health Research | PJT-180245 | D Peter Tieleman |
| Canadian Institutes of Health Research | PJT-175024 | David Fedida |
| Natural Sciences and Engineering Research Council of Canada | RGPIN-2022-03021 | David Fedida |
| Canada Research Chairs | | D Peter Tieleman |

The funders had no role in study design, data collection and interpretation, or the decision to submit the work for publication.

### Author contributions

Alessia Golluscio, Conceptualization, Data curation, Formal analysis, Investigation, Writing – original draft, Writing – review and editing; Jodene Eldstrom, Data curation, Formal analysis, Writing – original draft, Writing – review and editing; Jessica J Jowais, Kevin Peter Cunningham, Alicia De La Cruz, Xiaoan Wu, Data curation, Formal analysis, Investigation; Marta Elena Perez, Investigation; Valentina Corradi, Data curation, Formal analysis, Investigation, Writing – original draft, Writing – review and editing; D Peter Tieleman, Resources, Formal analysis, Supervision, Writing – original draft, Writing – review and editing; David Fedida, Formal analysis, Supervision, Writing – original draft, Writing – review and editing; H Peter Larsson, Formal analysis, Supervision, Writing – original draft, Project administration, Writing – review and editing

### Author ORCIDs

Alessia Golluscio ![ORCID] https://orcid.org/0000-0003-4556-9869
Kevin Peter Cunningham ![ORCID] http://orcid.org/0000-0001-6772-0107
Alicia De La Cruz ![ORCID] https://orcid.org/0000-0002-0349-4881
D Peter Tieleman ![ORCID] https://orcid.org/0000-0001-5507-0688
David Fedida ![ORCID] https://orcid.org/0000-0001-6797-5185
H Peter Larsson ![ORCID] https://orcid.org/0000-0002-1688-2525

Reviewer #1 (Public review): https://doi.org/10.7554/eLife.95852.4.sa1
Reviewer #2 (Public review): https://doi.org/10.7554/eLife.95852.4.sa2
Reviewer #3 (Public review): https://doi.org/10.7554/eLife.95852.4.sa3
Author response https://doi.org/10.7554/eLife.95852.4.sa4

## Additional files

### Supplementary files
• MDAR checklist

### Data availability

All data generated or analysed during this study are included in the manuscript and supporting files; source data files have been provided for *Figures 1, 3, 5 and 6*, *Figure 3—figure supplement 1*, *Figure 5—figure supplements 1 and 2*. *Figure 1—source data 1*, *Figure 3—source data 1*, *Figure 5—source data 1*, *Figure 6—source data 1*, *Figure 3—figure supplement 1—source data 1*, *Figure*

*5—figure supplement 1—source data 1*, *Figure 5—figure supplement 2—source data 1* contain the numerical data used to generate these figures.

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
