## [Editor Report · eLife assessment]

This article reveals an **important** mechanism of KCNQ1/IKs channel gating and PUFA modulation of this mechanism. This mechanism is supported by **convincing** single-channel recordings, macroscopic current recordings, and mutational analyses. These findings are of importance to the ion channel field and possibly future therapeutic applications.

---

## [Referee Report · Reviewer #1 (Public review)]

This study comes to an interesting conclusion: a polyunsaturated fatty acid, Lin-Glycine, increases the conductance of KCNQ1/KCNE1 channels by stabilizing a state of the selectivity filter that allows K+ conduction. The stabilization of a conducting state is well supported by single channel analysis, which shows that normally infrequent opening bursts occur more often in the presence of the PUFA. The linkage to PUFA action through the selectivity filter is supported by disruption of PUFA effects by mutation of residues which change conformation in two KCNQ1 structures from the literature. A definitive functional experiment is conducted by single channel recordings with selectivity filter domain mutation Y315F which ablates the Lin-Glycine effect on Gmax. The computational exploration of two selectivity filter structures proposed to interact distinctly with Lin-Glycine is informative. Both mutation results and simulations converge on the proposed selectivity filter mechanism, although other possibilities for Lin-Glycine binding and action might be possible. Overall, the major claim of the abstract is well-supported: "... that the selectivity filter in KCNQ1 is normally unstable ... and that the PUFA-induced increase in Gmax is caused by a stabilization of the selectivity filter in an open-conductive state."

---

## [Referee Report · Reviewer #2 (Public review)]

Golluscio et al. address one of the mechanisms of IKs (KCNQ1/KCNE1) channel upregulation by polyunsaturated fatty acids (PUFAs). PUFAs are known to upregulate KCNQ1 and KCNQ1/KCNE1 channels through two mechanisms: one shifts the voltage dependence in a negative direction, and the other increases the maximum conductance (Gmax). While the first mechanism is known to affect the voltage sensor equilibrium through a charge effect, the second mechanism is less understood. Using single-channel recordings and mutagenesis at putative PUFA binding sites, they successfully demonstrate that the selectivity filter is stabilized in a conducting state by PUFA binding, and that this is the mechanism by which PUFAs increase Gmax. Their single-channel recordings are straightforward and clearly show that the selectivity filter tends to become conductive upon PUFA binding. Since PUFAs are potential therapeutic reagents for cardiac arrhythmias such as long QT syndrome, their findings are beneficial for future research and applications of these compounds.

---

## [Referee Report · Reviewer #3 (Public review)]

Summary:

This manuscript reveals an important mechanism of KCNQ1/IKs channel gating such that the open state of the pore is unstable and undergoes intermittent closed and open conformations. PUFA enhances the maximum open probability of IKs by binding to a crevice adjacent to the pore and stabilize the open conformation. This mechanism is supported by convincing single channel recordings that show empty and open channel traces and the ratio of such traces is affected by PUFA. In addition, mutations of the pore residues alter PUFA effects, convincingly supporting that PUFA alters the interactions among these pore residues.

Strengths:

The data are of high quality and the description is clear.

---

## [Author Response]

The following is the authors’ response to the previous reviews.

**Reviewer #1 (Recommendations For The Authors):**
The additional data included in this revision nicely strengthens the major claim.I apologize that my comment about K+ concentration in the prior review was unclear. The cryoEM structure of KCNQ1 with S4 in the resting state was obtained with lowered K+ relative to the active state. Throughout the results and discussion it seems implied that the change in voltage sensor state is somehow causative of the change in selectivity filter state while the paper that identified the structures attributes the change in selectivity filter state not to voltage sensors, but to the change in [K+] between the 2 structures. Unless there is a flaw in my understanding of the conditions in which the selectivity filter structures used in modeling were generated, it seems misleading to ignore the change in [K+] when referring to the activated vs resting or up vs down structures. My understanding is that the closed conformation adopted in the resting/low [K+] is similar to that observed in low [K+] previously and is more commonly associated with [K+]-dependent inactivation, not resulting from voltage sensor deactivation as implied here. The original article presenting the low [K+] structure also suggests this. When discussing conformational changes in the selectivity filter, I strongly suggest referring to these structures as activated/high [K+] vs resting/low [K+] or something similar, as the [K+] concentration is a salient variable.

There seems to be some major confusion here and we will try to explain how we think. Note that in the Mandela and MacKinnon paper, there is no significant difference in the amino acid positions in the selectivity filter between low and high K+ when S4 is in the activated position (See Mandala and Mackinnon, PNAS Suppl. Fig S5 C and D). There are only fewer K+ in the selectivity filter in low K+. So, the structure with the distorted selectivity filter is not due to low K+ by itself. Note that there is no real difference between macroscopic currents recorded in low and high K+ solutions (except what is expected from changes in driving force) for KCNQ1/KCNE1 channels (Larsen et al., Bioph J 2011), suggesting that low K+ do not promote the non-conductive state (Figure 1). We now include a section in the Discussion about high/low K+ in the structures and the absence of effects of K+ on the function of KCNQ1/KCNE1 channels.

**Author response image 1. sa4fig1:** Macroscopic KCNQ1/KCNE1 currents recorded in different K+ conditions. Note that there is no difference between current recorded in low K+ (2 mM) conditions and high (96 mM) K+ conditions (n = 3 oocytes). Currents were normalized in respect to high K+.

Note also that, in the previous version of the manuscript, we did not propose that the position of S4 is what determines the state of the selectivity filter. We only reported that the CryoEM structure with S4 resting shows a distorted selectivity filter. It seems like our text confused the reviewer to think that we proposed that S4 determines the state of the selectivity filter, when we did not propose this earlier. We previously did not want to speculate too much about this, but we have now included a section in the Discussion to make our view clear in light of the confusion of the reviewers.

It is clear from our data that the majority of sweeps are empty (which we assume is with S4 up), suggesting that the selectivity filter can be (and is in the majority of sweeps) in the non-conducting state even with S4 up. We think that the selectivity filter switches between a non-conductive and a conductive conformation both with S4 down and with S4 up. The cryoEM structure in low K+ and S4 down just happened to catch the non-conductive state of the selectivity filter. We have now added a section in the Discussion to clarify all this and explain how we think it works.

However, S4 in the active conformation seems to stabilize the conductive conformation of the selectivity filter, because during long pulses the channel seems to stay open once opened (See Suppl Fig S2). So, one possibility is that the selectivity filter goes more readily into the non-conductive state when S4 is down (and maybe, or not, low K+ plays a role) and then when S4 moves up the selectivity filter sometimes recovers into the conductive state and stays there. We now have included a section in the Discussion to present our view. Since this whole discussion was initiated and pushed by the reviewer, we hope that the reviewers will not demand more data to support these ideas. We think that this addition makes sense since other readers might have the same questions and ideas as the reviewer, and we would like to prevent any confusion about this topic.

Figure 1It remains unclear in the manuscript itself what "control" refers to. Are control patched the same patches that later receive LG?

Yes, the control means the same patch before LG. We now indicate that in legends and text throughout.

Supplementary Figure S1Unclear if any changes occur after addition of LG in left panel and if the LG data on right is paired in any way to data on left.

Yes, in all cases the left and right panel in all figures are from the same patch. We now indicate that in legends and text throughout.

The letter p is used both to represent open probability open probability from the all-point amplitude histogram and as a p-value statistical probability indicator sometime lower case, sometimes upper case. This was confusing.

We have now exclusively use lower case p for statistical probability and Po for open probability.

"This indicates that mutations of residues in the more intracellular region of the selectivity filter do not affect the Gmax increases and that the interactions that stabilize the channel involve only residues located near the external region part of the selectivity filter. "Seems too strongly worded, it remains possible that mutations of other residues in the more intracellular region of the selectivity filter could affect the Gmax increases.

We have changed the text to: "Mutations of residues in the more intracellular region of the selectivity filter do not affect the Gmax increases, as if the interactions that stabilize the channel involve residues located near the external region part of the selectivity filter. "

Supplementary Figure S7Please report Boltzmann fit parameters. What are "normalized" uA?

We removed the uA, which was mistakenly inserted. The lines in the graphs are just lines connecting the dots and not Boltzmann fits, since we don’t have saturating curves in all panels to make unique fits.

"We have previously shown that the effects of PUFAs on IKs channels involve the binding of PUFAs to two independent sites." Was binding to the sites actually shown? Suggest changing to: "We have previously proposed models in which the effects of PUFAs..."

We have now changed this as the Reviewer suggested: " We have previously proposed models in which the effects of PUFAs on IKs channels involve the binding of PUFAs to two independent sites."

Statistics used not always clear. Methods refer to multiple statistical tests but it is not clear which is used when.

We use two different tests and it is now explained in figure legends when either was used.

n values confusing. Sometimes # of sweeps used as n. Sometimes # patches used as n. In one instance "The average current during the single channel sweeps was increased by 2.3 {plus minus} 0.33 times (n = 4 patches, p = 0.0006)" ...this sems a low p value for this n = 4 sample?

We have now more clearly indicated what n stands for in each case. There was an extra 0 in the p value, so now it is p = 0.006. Thanks for catching that error.

**Reviewer #2 (Recommendations For The Authors):**
I still have some comments for the revised manuscript.(1) (From the previous minor point #6) Since D317E and T309S did not show statistical significance in Figure 5A, the sentences such as "This data shows that Y315 and D317 are necessary for the ability of Lin-Glycine to increase Gmax" or "the effect of Lin-Glycine on Gmax of the KCNQ1/KCNE1 mutant was noticeably reduced compared to the WT channel showing the this residue contributes to the Gmax effect (Figure 5A)." may need to be toned down. Alternatively, I suggest the authors refer to Supplementary Figure S7 to confirm that Y315 and D317 are critical for increasing Gmax.

We have redone the analysis and statistical evaluation in Fig 5. We no use the more appropriate value of the fitted Gmax (which use the whole dose response curve instead of only the 20 mM value) in the statistical evaluation and now Y315F and D317E are statistically different from wt.

(2) Supplementary Fig. S1. All control diary plots include the green arrows to indicate the timing of lin-glycine (LG) application. It is a bit confusing why they are included. Is it to show that LG application did not have an immediate effect? Are the LG-free plots not available?

Not sure what the Reviewer is asking about? In the previous review round the Reviewers asked specifically for this. The arrow shows when LG was applied and the plot on the right shows the effect of LG from the same patch.

(3) The legend to Supplementary Figure S4, "The side chain of residues ... are highlighted as sticks and colored based on the atomic displacement values, from white to blue to red on a scale of 0 to 9 Å." They look mostly blue (or light blue). Which one is colored white? It might be better to use a different color code. It would also be nice to link the color code to the colors of Supplementary Figure S5, which currently uses a single color.

We have removed “from white to blue to red on a scale of 0 to 9 Å” and instead now include a color scale directly in Fig S4 to show how much each atom moved based on the color.

We feel it is not necessary to include color in Fig S5 since the scale of how much each atom moves is shown on the y axis.

(4) Add unit (pA) to the y-axis of Supplementary Figure S2.

pA has been added.

**Reviewer #3 (Recommendations For The Authors):**
Some issues on how data support conclusions are identified. Further justifications are suggested.186: “The decrease in first latency is most likely due to an effect of Lin-Glycine on Site I in the VSD and related to the shift in voltage dependence caused by Lin-Glycine." The results in Fig S1B do not seem to support this statement since the mutation Y315F in the pore helix seemed to have eliminated the effect of Lin-Glycine in reducing first latency. The authors may want to show that a mutation that eliminating Site I would eliminate the effect of Lin-Glycine on first latency. On the other hand, it will be also interesting to examine if another pore mutation, such as P320L (Fig 5) also reduce the effect of Lin-Glycine on first latency.

These experiments are very hard and laborious, and we feel these are outside the scope of this paper which focuses on Site II and the mechanism of increasing Gmax. Further studies of the voltage shift and latency will have to be for a future study.

The mutation D317E did not affect the effect of Lin-Glycine on Gmax significantly (Fig 5A, and Fig S7F comparing with Fig S7A), but the authors conclude that D317 is important for Lin-Glycine association. This conclusion needs a better justification.

We have redone the analysis and statistical evaluation in Fig 5. We no use the more appropriate value of the fitted Gmax (which use the whole dose response curve instead of only the 20 mM value) in the statistical evaluation and now D317E is statistically different from wt